# Understanding Working Memory and Mathematics Development in Ethnically/Racially Minoritized Children through an Integrative Theory Lens

**DOI:** 10.3390/bs14050390

**Published:** 2024-05-06

**Authors:** Dana Miller-Cotto, Andrew D. Ribner, Leann Smith

**Affiliations:** 1Department of Psychological Sciences, 600 Hilltop Drive, Kent State University, Kent, OH 44240, USA; 2Department of Psychology, 7 Woodland Road, Pittsburgh, PA 15232, USA; 3Department of Educational Psychology, Harrington Education Center Office Tower, 540 Ross St Suite 705, Texas A&M University, College Station, TX 77843, USA

**Keywords:** social capital theory, community cultural wealth, working memory, mathematics, ethnic/racial minority children

## Abstract

Limited research on working memory has centered on ethnically/racially minoritized children, thereby limiting researchers’ abilities to draw conclusions about working memory or to provide additional supports in cultivating working memory for these children. Using the Integrative Theory as a lens, the current study explored the predictive benefit of parent academic socialization strategies on the working memory and subsequent mathematics skills of a nationally representative sample of ethnically/racially minoritized children. Using structural equation modeling techniques, a path model including social position; family structure; leisure activities; parent academic socialization strategies; and their association with kindergarten Asian/Asian-American (*N* = 1211), Black (*N* = 1927), and Latine (*N* = 3671) children’s working memory and first-grade mathematics skills were examined. Furthermore, multigroup moderation was used to test for differences between ethnic/racial groups. Connections to social capital theory, community cultural wealth, and culturally relevant interpretations of the study findings are discussed.

## 1. Introduction

Despite predictions of ethnic/racial minorities becoming the majority in 2045 [1], schools still fail to produce conditions in which ethnically/racially minoritized children can thrive. In fact, only 40% of Black and Latine children frequently demonstrate mathematics proficiency by the fourth grade [2]. One variable that is a strong predictor of mathematical proficiency is working memory (WM), a processing resource of limited capacity involved in preserving information while simultaneously processing the same or other information [3]. Unfortunately, Black, Latine, and/or children living in poverty often perform worse on measures of WM than do their White, high-income peers [4]. Past research has suggested that this performance is likely due to poor parenting, household chaos, and poverty [5,6]. Unfortunately, this assumption focuses on perceived problems rather than the contexts in which children are reared and assumes a one-size-fits-all model of WM development [4] and the role of WM in the development of mathematical skills [3]. Indeed, there is a long history of decontextualizing ethnic/racial minority children’s experiences by using a deficit-based lens, concluding that ethnic/racial minority children are lacking because their environments are unlike their White upper middle-class counterparts culturally. However, some work has suggested that ethnic/racial minority children are socialized with a set of values and traditions by their parents to thrive in the U.S. despite it being a context riddled with systemic and interpersonal discrimination, negatively affecting their development [7]. In particular, attempts have been made to understand parenting and home environment outside the context of the larger ecosystem and sociopolitical histories, blaming distal factors for poor working memory instead of more proximal, and appropriate, areas of influence [4]. An area ripe for understanding minoritized children’s development can be found by studying family practices, or assets, that may have emerged because of this larger ecosystem [8].

Here, we aim to challenge this one-size-fits-all approach to studying early cognitive development. We frame the current work using insights from Integrative Theory [7] combined with familial capital [9] and a community cultural wealth [10] perspective to understand the various factors influencing the academic performance of racially/ethnically minoritized children in the U.S. In the current study, we used this framework to examine what home context variables (e.g., parent socialization practices or family practices and home experiences) predict individual differences in performance in early mathematics and WM tasks between and within these ethnic/racial minority groups.

## 2. Historical Context of Children of Color in the U.S.

In the United States, ethnic/racial minority children comprised 50% of births in 2011 [1]. The term *minority children and families* refers to individuals from various non-White racial groups and people from a particular region of the world or country who share characteristics such as culture, language, or beliefs [11]. For many ethnic/racial minority children, despite making up a significant portion of children in the U.S., educational outcomes are grim relative to White children [12]. Parental experiences with racial discrimination and consequent stress can be passed on to their children through various messages, potentially influencing academic outcomes [13,14,15]. However, it is essential to acknowledge that minoritized children and their families have noteworthy cultural histories as they relate to their sociopolitical positions in the U.S. [7,16]. These groups also have their own cultural assets, the set of values and beliefs that minority families use to raise healthy children [7,17]. Evidently, these groups have employed various practices related to cultural assets within their families and communities to thrive in the U.S., employing familial capital (e.g., extended family members) and academic socialization practices (e.g., engaging in leisure activities, messages about the importance of school, and racial socialization discussions). Why might we expect different effects of academic socialization between different ethnic/racial groups? One possibility is that Black, Latine, and Asian/Asian-American children, due to different histories of migration in the U.S., may have developed different practices and home environments in response to these reasons for migration. Families who fled war and political strife are likely to have developed resilience strategies that differ from families who have been in the U.S. for several generations via involuntary migration (i.e., the trans-Atlantic slave trade). Indeed, in one meta-analysis [15], relations between racial socialization, or messages about race, and academic outcomes were stronger and positive for Black children relative to their Latine counterparts, who had a positive but weaker relationship, and Asian/Asian-American peers, who demonstrated a negative relationship. Thus, it may be the case that, for some families, these processes are not predictive of academic outcomes. To frame this discussion further, we combine several theoretical perspectives: we present the Integrative Theory [7] at the forefront, in combination with familial capital, and a community cultural wealth perspective [9], described in greater detail in the next section.

### Multi-Theoretical Framework

It has been long believed that development is a product of an interaction between the individual and their environment through the course of their lifetime [18]. This includes their home, neighborhood, and various other communities that may exert influence on their lives. Work using Bronfenbrenner’s Ecological Systems theory [18] as it relates to the home environment is abundant but not sufficient alone in understanding the development of ethnic/racial minority children in the United States. Alternatively, Integrative Theory [7] takes a different approach to understanding developmental processes as they relate to ethnic minority children. Generally, parents of all ethnicities hold attitudes and employ specific socialization processes that are designed to influence children’s academic expectations, values, and behaviors [19,20]. Academic values are often developed through implicit or explicit messages within the home environment and accompany children as they enter school, often shaping children’s performance in the learning process [21,22]. The Integrative Theory focuses on several aspects of development through family processes that may promote learning and academic outcomes specifically for ethnic minority children [23,24]. One component, cultural adaptation, is resilience-based, focusing on how families adapt to a culture through values and beliefs (e.g., child-rearing values and attitudes) and behaviors (e.g., language use and parenting practices) that differ from the dominant culture [25]. Cultural adaptation is developed and socialized via a family context and is relevant for first and later generations of immigrant parents and children [26]. Further, family traditions and routines that develop through cultural adaptation may help us better understand how ethnic minority children and their families develop practices to thrive in the U.S. [7,27]. Family practices have rarely been framed this way in developmental psychology, particularly as it relates to ethnic/racial minority children’s cognitive development. Thus, we attempt to understand these practices herein.

Parents of children from ethnic/racial minority backgrounds are aware of the challenges their children will face as they navigate life in the U.S. [7]. To circumvent these challenges, these parents leverage various forms of capital, including familial capital. Familial capital may include the transmission of traditions of values favoring the community and bonds within the community that students can then bring into the classroom with them [9]. Previous studies have shown the linkage between a strong belief in the value of familism and the educational and career success of the children of Latine immigrant parents [28] and high levels of academic motivation for learning [29] among Latine adolescents. Toyokawa and Toyokawa [30] examined the role of familism, the obligation to one’s family, hypothesizing that greater family cohesion should be related to better academic outcomes. The findings indicated that family cohesion was positively associated with adolescents’ GPA and educational expectations; family obligation, on the other hand, was negatively associated with academic outcomes for Latine youth. The distinction between family obligations versus family cohesion is important. If youth believe they have extra responsibilities placed on them, they may prioritize family obligations over schooling. However, if they feel like they belong to their families, they can rely on them for emotional support in the face of adversity; this is likely positively related to academic outcomes. Family cohesion, therefore, may serve as a familial capital function. Family capital can come in the form of the inclusion of extended family members in providing support, the transmission of messages and lessons, and access to opportunities that promote academic outcomes. For ethnic/racial minority families, racial socialization messages often include messages from parents articulating values and perspectives about ethnicity and race to their children, which are often deemed a cultural asset for children of color that buffers against racist encounters and aids in the effective processing of stereotypes and microaggressions [16] Indeed, these practices have been associated with promising academic outcomes for minoritized children [15]. Further, the combination of familial support, academic socialization, and engagement in leisure activities has been another resource families of color leverage to have autonomy over their children’s outcomes. These various forms of familial capital are indeed important for a child’s intellectual development. This may raise the question, what family practices promote school readiness skills specifically, such as WM? And if there are any that do, do they extend to math skills? Here, we aim to understand these practices and their associations with school readiness skills like WM and mathematics.

## 3. Home Environment

*Family Structure and Connectedness*. In many families, parents are not the only adults to assist with child-rearing; indeed, older siblings and grandparents serve as additional support [31]. In one study, grandparents and older siblings spent more time with children engaging in learning activities as opposed to parents [32]. Despite that study taking place in South Africa, many ethnically/racially minoritized families in the U.S. have similar family structures in the home, relying on extended family for support and child-rearing. Thus, it is reasonable to include these variables in our investigation.

Family members who can depend on one another, either for emotional or financial support, are related to enhanced academic performance. Indeed, Bravo and colleagues [28] have demonstrated that there is a strong relationship between obligation to one’s family and educational outcomes for Latine children. This may have positive long-term implications for Latine children. Fuligni [29] found that this obligation was also related to academic motivation among Latine adolescents. More recent work has also supported this notion, suggesting that family obligation is positively associated with adolescents’ grades and educational expectations overall [30]. Alternatively, Latine family obligation, in this case, familism, while playing an important role in Latin culture, may also negatively affect academic outcomes. For instance, when children must serve as language brokers for adults who may not speak the dominant language in a given context, this can pose a greater responsibility to these children.

Similar patterns are demonstrated for other ethnic and racial groups. Parents from Asian, Black, and Latine families who have higher educational expectations and voice the importance of academics is associated with enhanced academic outcomes [12]. Family relationships may play a role in broader cognitive development. However, it is unclear how these family structures may support cognitive development and subsequent academic outcomes. Here, we aim to determine these relationships concurrently and longitudinally and how these might differ across ethnic/racial groups.

Ethnic minority children and their families seem to draw from different values and strengths within their culture, community, and family [33]. Asian American families, particularly those from East Asia (China, Japan, Korea, and Taiwan), seem to value collectivism, interdependence, prioritization of the group over self, respect for elders, and strong achievement orientation [17,34]. Black families value racial socialization [35,36], positive identity development [37], and communalism [38]. Latine families value familism or family cohesion, and respeto (i.e., respect for elders and self, moral education, and interpersonal relationships [33]. All groups have high expectations for their children to succeed [12,39]. Thus, this is not to say that there is no overlap between these groups and what they value, but rather, the prevalence or kinds of discussions around these may vary from group to group within families. This lens is useful for understanding what familial capital children of ethnic/racial minority families pull from to thrive academically [10]. Here, we aim to determine these relationships concurrently and longitudinally and how these might differ across ethnic/racial groups.

One important consideration when comparing these three ethnic minority groups is that, historically, Asian, Black, and Latine families have been studied using different terms (e.g., acculturation, ethnic–racial socializations, family support, and familism). Though neither familism nor acculturation is the focus of the current study, they support the notion that family support, through a range of factors, may be an important mechanism through which academic development occurs. We use family practices as a broader term to capture the similarities across cultures while also acknowledging there are indeed cultural differences between these groups. We attempt to determine specific aspects of family practices within the household to better understand these processes and how they might be related to academic performance across ethnic/racial minority children.

*Family Routines and Leisure Activities*. While some work has suggested that parent–child interactions are associated with academic skills and WM [40,41], there is less known about family practices as they relate to both constructs. Some qualitative work has noted the importance of leisurely activities and parental academic activities with children in academic outcomes [42]. Regarding mathematics skills more specifically, some work suggests that number talk, playing mathematics games [43], or using mathematically based language at home is positively related to number knowledge in kindergarten [44] with some work suggesting that number talk with quantities larger than three predict children’s later cardinality ability [45]. In Latine families, one study [46] found that food routines such as grocery shopping and/or measuring and cooking together were associated with children’s mathematical outcomes. Understanding specific family routines, whether related or unrelated to specific domains, may be an important aspect of the family home life in leveraging learning opportunities for ethnic minority children.

*Parental Warmth*. Parental practices have been related to WM and executive functions more broadly, an umbrella term to which WM belongs. Much of this work is exclusively centered around types of parental attachment. For instance, Sosic [6] investigated the relationship between various aspects of parenting and children’s executive function skills. These aspects included positive involvement, supervision, monitoring, positive discipline, consistency with discipline, the use of corporal punishment, and authoritarian parenting. For children ages 9 to 14 in that study, the results suggested that high parent involvement was broadly associated with better executive function skills, regardless of SES. These associations appear to be rooted early in life: one study found that parents transmit their own EF skills to their young children via such practices as autonomy-supportive behavior (i.e., providing scaffolding in goal-directed activities in responsive ways that provide an optimal level of challenge) and sensitivity [47]. Further, sensitive, autonomy-supportive, and cognitively stimulating parenting at 6 months predicts better attentional shifting at 36 months [48]. Harsh parenting attitudes have been found to be negatively associated with inhibitory control and WM [49]. We focus on family practices that may facilitate WM and mathematics skills.

## 4. Parent Academic Socialization

Parent academic socialization is a multidimensional construct that describes the ways parents pass down values and expectations, highlighting the importance of school and the usefulness of academic preparedness for future life success [50]. Academic socialization practices may include but are not limited to home-based academic support and parental–child activities, as well as the expressed beliefs and values related to academic achievement. Prior work has shown that academic socialization promotes greater academic achievement [51] even when considering peer influences [52]. While prior research has demonstrated the significant positive role of these factors in mathematics ability [12,53], it is unclear how parental academic socialization predicts WM development, though some work may suggest that ethnic/racial minority families make children aware of self-regulation to do well in school. One possibility could be that, for many Latine families, being *bien educado* requires that their children also behave and have good manners, which requires self-regulation [10]. Here, we examine parental expectations, parent academic socialization, including racial socialization or messages about and pertaining to race and social position within society, and their roles in WM and mathematics ability.

## 5. Family Practices, Working Memory, and Mathematics

As previously noted, some work has suggested that parent–child interactions are associated with academic skills and working memory [40,41], there is less known about family practices as they relate to both constructs, with qualitative work demonstrating the importance of leisurely activities and parental academic activities with their children in academic outcomes [42]. 

Another aspect of school readiness, executive functions (EFs), a set of processes related to cognitive control related to goal-directed behavior to which WM belongs [54], is frequently associated with academic performance [55]. While it is well established that EFs are heritable [56], much less is known about how the environment, namely, family structures and practices, plays a role in EFs for minoritized children. At the time of this writing, some work suggests that children living in high-poverty neighborhoods demonstrate poorer EFs than their peers in higher socioeconomic status (SES) homes, though enrollment in school helps to reduce inequities in opportunities related to growth in EFs [57]. However, the underlying mechanisms are yet to be fully understood. Much of this work is exclusively centered around types of parental attachment. For instance, Sosic and colleagues [6] found that high parent involvement was broadly associated with better EF skills, regardless of SES. Further, household chaos, defined as confusion, ambient noise, and clutter around the home has been associated with EFs through parenting behaviors [58]. That is, a chaotic home often predicts parenting practices that negatively impact behavior regulation and EFs. Of course, these factors are highly tied to high-poverty homes, not specific to ethnic minority status, and difficult to disentangle from hardships associated with a lack of resources, and the term chaos may be an example of deficit framing. Indeed, many family practices and values may not map well to typical, Western cognitive tasks. Gaskins and Alcalá [23] suggest that neither measures nor the research protocols used to assess EF take into account the cultural values of non-White children, making interpretations difficult. For instance, they argue that many EF tasks are acontextual and include a series of instructions that have no cultural meaning to Yucatec children. Lower performance in these tasks is assumed to be reflective of these children’s lack of ability and is not due to the tasks being acontextual and not attached to any culturally meaningful goal or norm. Thus, we focus our attention on family practices that may facilitate learning and development opportunities, particularly as this relates to an aspect of EF skills, namely, WM.

## 6. The Current Study

We take a strengths-based perspective, calling on work from Integrative Theory [7] and the benefits of familial capital, to explore how ethnic/racial minority families within the U.S. make use of parental academic socialization strategies (expectations, routines, and leisure activities) to promote the development of mathematics and WM. Specifically, we explored family practices related to involvement (either via parents or older caregivers and siblings), stability (e.g., routines), and parental expectations across groups while controlling for social position variables to better understand these processes. We expand our study by not only examining practices as they relate to parents but all caregivers (e.g., grandparents and older siblings) as well, influenced by social capital theory but focusing primarily on familial capital. Given that community cultural wealth as it relates to social capital theory and Integrative Theory have specifically written about ethnic/racial minority children, we focus only on Asian, Black, and Latine children in this study; in addition, we use principles of Quantitative Critical Theory [59] in our analysis of patterns of associations for groups of children with different racial, cultural, and linguistic experiences in the U.S. This study addresses the following two research questions:Are parent academic socialization practices predictive of WM and mathematics ability for a nationally representative sample of ethnic/racial minority children at school entry?Does WM mediate the relationship between parent academic socialization practices and later mathematics ability for a nationally representative sample of ethnic/racial minority children?Are there differences in the prediction of parent academic socialization practices regarding mathematics ability and WM based on children’s ethnic minority membership?

If WM does not mediate the relationship, then a direct relationship between parenting factors and mathematics might offer some insights about parent practices for minoritized youth. Because these analyses are exploratory, we anticipate that the magnitude of these relationships by ethnic/racial minority group will be equivalent in most variables except for ones that have previously demonstrated differences [15].

## 7. Method

### 7.1. Sample

Participants were from the Early Childhood Longitudinal Study, Kindergarten Cohort-2011 (ECLS-K:2011), the third early childhood longitudinal study sponsored by the United States. A three-stage probability sampling design that involved stratified sampling, proportional sampling, and a cluster sampling strategy was used to achieve a nationally representative sample of 18,200 kindergarten children (rounding to the nearest 50 in accordance with the restricted data use license from the U.S. federal agency) enrolled in 968 public and private schools from all 50 states during the 2010–2011 school year. At each selected school, approximately 23 kindergarten children were randomly selected. Participating parents provided informed consent. State education officers, public school administrators, parochial school leaders, and private school administrators provided endorsement for the study prior to data collection. We received a restricted-use license from the United States federal agency to use the ECLS-K data for 2010–2011. In conducting a secondary data analysis of this dataset, the Institutional Review Board of the first author’s institution deemed this study exempt from review because of the exemption reason pertaining to the analysis of already collected data with no identifiers. Thus, because this was a secondary data analysis, we did not have access to consent forms for the study given the nature of the data. The data that support the findings of this study are available from the corresponding authors upon reasonable request.

This study focused on a subset of 6809 Asian, Black, and Latine children in the ELCS-K. Of those, there were 1211 Asian-American and Native Hawaiian children, 1927 Black children, and 3671 Latine children entering kindergarten for the first time in fall 2010. Across the groups, 17% to 45% of these children came from families from below the poverty threshold (defined by the U.S. government on a national level as an income guideline for administrative purposes related to receipt of government services) (*M* = 1.96, median = 2.00); 49% to 53% of these children identified as girls (*M* = 1.5, median = 1.00); and average age of children ranged from 72 to 73 months at kindergarten enrollment. Between 47% and 96% of these children spoke English as the primary language at home (*M* = 2.22, Median = 2.00).

Most ethnic/racial minority children in this study were born in the U.S. (83% Asian-American, 97% Black, and 96% Latine); however, only about 16% of Asian-American parents were born in the U.S., compared with 89% of Black and 44% of Latine parents. Less than half of the Asian-American families spoke English at home compared with 96% of Black families and 54% of Latine families. Due to the variability in where children and their families originated from, we opted to use the terms Asian-Americans to denote Asian-American and Pacific Islander (AAPI) children and their families currently living in the U.S.; Black to denote inclusion of African-American families, as well as descendants of Africans from the Caribbean and West Africa living in the U.S.; and Latine to denote families from Latin America living in the U.S.

Each of these ethnic/racial minority groups was diverse. The Asian-American families came from over 53 countries, including the United States (15.9%), India (20%), China (15.8%), Vietnam (12.1%), the Philippines (8.7%), South Korea (4.4%), Pakistan (3.5%), Taiwan (2.3%), and Thailand (1.9%). The Black families came from over 40 countries, including the United States (88.6%), Haiti (2.4%), Jamaica (1.5%), Somalia (1%), Nigeria (0.7%), and Sudan (0.7%). The Latine families came from over 50 countries, including the United States (43.5%), Mexico (39.3%), El Salvador (3.8%), Puerto Rico (1.9%), Guatemala (1.8%), Cuba (1.1%), Colombia (1%), Ecuador (1%), Honduras (1%), and the Dominican Republic (0.9%).

Finally, these ethnic/racial minority children also came from different regions of the United States. Around 45% of Asian-Americans came from the West and 22% from the Northeast; around 60% of Black children came from the South and 22% from the Midwest; and around 41% of Latine children came from the West and 36% from the South.

### 7.2. Measures

Direct assessment data were collected in the fall of the kindergarten year. Fall assessments were conducted from August through mid-December in a one-on-one session with data collectors in a quiet school setting; spring assessments were conducted from March through June. Some additional data come from parent interviews, the most comprehensive of which were completed in the first wave of data collection (fall of the kindergarten year); all parent-report data come from that time point.

*Mathematics skills*. The mathematics standardized assessment, measured during the end of the fall of kindergarten, first grade, and second grade, assessed children’s conceptual knowledge, procedural knowledge, and problem-solving within specific strands, with a focus on the “number sense, properties, and operations” strand. Item response theory modeling was used to create scores for each child that could be compared across time [60]. These scores were supplied in the ECLS-K dataset. Test items were designed to tap a broad range of skills that are typically taught, important skills for that grade, and skills that are consistent with national and state standards (e.g., NAEP, NCTM, etc.). Spanish versions of the assessment were given to children who did not pass the screener test in English. IRT modeling, which was performed by ECLS-K data researchers before being made available to the public and supplied in the ECLS-K dataset, was used to create scores that could be compared across time [60]. A two-stage adaptive test was used to measure these constructs, involving a routing test to ascertain where a student was located developmentally throughout a sequence of items, ending with a second stage of grade-appropriate items. This was performed to ensure that a student did not take all items while still obtaining an accurate score (alpha ranged from 0.91 to 0.94).

*Working memory*. WM was measured using the numbers reversed subtest of the Woodcock-Johnson III Tests of Cognitive Abilities, where children are asked to repeat a series of digits in reverse sequence. The strings of digits became progressively longer (up to eight digits) until the child recited three consecutive number sequences incorrectly. For each child, a W score was computed to allow for modeling growth over time. A W score of 393 was the baseline value for children who were unable to answer any items correctly according to the ECLS-K:2011. WM assessments were also administered in Spanish for children whose home language was Spanish. Reliability was greater than 0.90. It was assessed during the fall of kindergarten and the fall of first grade in the current analyses.

### 7.3. Independent Variables

Variables were categorized as being a component of social position or family structure. Social position was measured by family socioeconomic status (SES), mother’s education level (1 = high school or higher; 0 = no high school diploma), and language. Family SES was a composite variable derived from parent responses to questions about family income and parental education. Language was coded based on whether the primary language spoken in the home was English or bilingual (1) compared with solely a language other than English (0). Family structure was measured by parent-reported maternal marital status (1 = married; 0 = not married), number of close grandparents (ranging from 0 to 5), and number of siblings in the household (ranging from 0 to 9). Child characteristics were also important to include. Parent-reported child age in months at the end of kindergarten, parent-reported child health (reverse-coded as 1 = poor, 2 = fair, 3 = good, 4 = very good, and 5 = excellent), and child gender (which was coded as a dummy variable, with “girl” coded as 1 as the referent), were all included in analyses. Finally, scores from a direct assessment of child reading skills at kindergarten entry—a correlate of both mathematical skills and WM—were included as a covariate.

## 8. Parent Academic Socialization Practices

All measures of parent academic socialization practices came from the parent interviews, which were conducted in the fall and spring of kindergarten. A trained interviewer phoned the parent at his or her home and conducted a 45–50 min interview or conducted the interview in person. Computer-assisted interviewing methods were used to record the parents’ answers.

We operationalized parental expectations (e.g., beliefs minority parents have about their child’s achievement capabilities, as well as the available resources for supporting a given level of achievement), family routines (e.g., the ways minority families organize their collective lives around daily living routines), and leisure activities (e.g., the ways minority children spend time on discretionary leisure activities by themselves or with their families) in the following ways. Parents reported what degree they expected of their children out of seven options (ranging from receiving less than high school to finishing a Ph.D., M.D., or other advanced degrees). As we are interested in measuring high parental expectations, this variable was coded 1 if parents expected their child to at least graduate from a four- or five-year college degree and coded 0 if parents had a lower degree of expectations for their child. This threshold was selected to be comparable to current research on parental expectations for students of color [15] Parental warmth was assessed during parental interviews at time point 1, where parents were asked questions regarding discipline, warmth, and emotional supportiveness. Finally, racial socialization was taken from a parental interview at time point 1, where parents were asked questions regarding beliefs about race and expectations for their children. Information regarding this interview is available in the ECLS-K user’s manual.

Promoting environments consisted of parent reports on what bedtime routine, that is, what time the child usually goes to bed. We coded the sleep routine variable as 1 if parents reported that their child usually went to bed by 8 p.m. and 0 if parents reported a later sleep time. This served as a proxy for measuring whether there were routines to support the child receiving the recommended 10 to 12 h of sleep like prior research that used indicators of wake-up times to demonstrate routines [61] Parents also reported on whether there were family rules for when their child ate meals and about how early or late the child watched television: TV time limits. As current research on sleep and child development also recommends family routines that monitor the use of screen time, including television watching [62] we coded this variable as 1 if there were family rules on how early or late the child may watch television and 0 if there were no family rules.

Leisure activities were measured by several variables based on parent-reported activities that they performed with their child. Parents reported whether the child participated in organized sports or physical structured activities. Furthermore, we included whether parents or any other family members frequently played games and puzzles with the child (coded as 1 = played three to seven times a week; 0 = less than three times a week), whether the child participated in music lessons or nonphysical structured activities, and whether they received tutoring (1 = yes; 0 = no). Parent-to-child and child self-reading activities were both measured, as prior research has found leisure reading at home to be highly predictive of child ability [53]. Parents were asked, in a typical week, how often they or any other family members read books to their child (coded as 1 = not at all, 2 = once or twice a week, 3 = three to six times a week, and 4 = every day). Parents were also asked to report, in the past week, how often children read to or pretend to read to themselves or to others outside of school (1 = not at all, 2 = once or twice a week, 3 = three to six times a week, and 4 = every day). Parents were also asked to report on whether their children engaged in computer activities (yes = 1; no = 0), computer activities, and whether parents or any family members took the child to visit the public library, museums, or the zoo (all coded as 1 = yes; 0 = no): outings.

### Analytic Approach

Integrative Theory for the development of minority children [7] argues that, when studying ethnic minority children, it is necessary to understand them based on their context and the ways in which they and their families may attempt to navigate as marginalized groups in society. Prior to the integrative model, most models that examined the developmental processes of minority children through a deficit model; White children were considered the standard, and any differences related to ethnic minority children were based on deficiencies [24]. One key developmental domain that is adversely impacted by ethnic–racial subordination is academic outcomes [7]. Of course, the home environment is embedded within a macro-structure; the home is within a neighborhood, influenced by social markers that may influence access to resources for children and their families. Garcia Coll and colleagues [7] argue that this can include housing segregation, access to quality health care, and the type of school children can attend in their neighborhoods, which are all part of environments that can promote or inhibit aspects of a child’s development and outcomes. Consequently, under these conditions, families with ethnic minority children develop approaches that will help their children thrive under these adverse conditions. Rather, the model focuses on how social position for children and their families informs conditions that characterize these families and takes a resilience perspective. In this way, children and their families employ an adaptive culture by developing a set of goals, values, and attitudes that inherently differ from the dominant (i.e., White American) culture to help children express agency in the face of this marginalization [7,24]. These adaptive behaviors are socialized through the family process—including aspects of adaptive cultures; promoting environments; and leisure activities that promote group harmony—a cultural value common across many ethnic/racial groups. However, one important note is that, while this has helped clarify developmental processes for ethnic minority children, little work has been performed to explicate the directions of effects within the model. Perez-Brena and colleagues [24] argue that aspects of the original model are indeed expected to interact with each other, as well as some mediational processes, and perhaps even some reciprocal processes, but little has been explored in this direction. Thus, in the current study, we explore these relationships using the ECLS-K dataset.

Analyses were conducted in several stages, with structural equation modeling (SEM) and multigroup moderation analyses being used to address the primary research questions. Prior to addressing the specific research questions, correlational analyses were conducted to ensure that independent variables were related to dependent variables. To answer research question 1 (are parent academic socialization practices predictive of mathematics and WM for a nationally representative sample of minority children?), SEM techniques for a just-identified path model were used. An inspection of regression paths will provide insight into specific relationships among variables. For the second research question, which explores the mediation of WM in the prediction of parent academic socialization regarding mathematics for a sample of ethnically diverse youth, indirect and residual direct effects were estimated with 500 bootstraps. For the final research question, which explores whether differences in the prediction of mathematics ability and WM based on parent academic socialization are found based on children’s ethnic/racial minority group membership, multigroup moderation was conducted with Asian, Black, and Latine subsamples.

MPlus was used for all analyses, which enabled the use of full information maximum likelihood (FIML) estimation to correct for missingness in SEM analyses. All exogenous variables were allowed to correlate to invoke FIML estimation. Data were missing on an average of 18% of cases and were determined to be missing at random (MAR). That is, missingness was not predicted by any demographic or performance variables that might explain our outcomes. Missingness for focal analyses was corrected using FIML through MPlus to maximize sample size and reduce possible bias in estimates [63]. For multigroup moderation analyses, multiple imputation with 10 datasets in MPlus was used.

## 9. Results

### 9.1. Descriptive Statistics

Descriptive information on the sample and bivariate correlations among variables in the model can be found in Table 1. In terms of social positions, Latine children, on average, came from the poorest families, followed by Black children and then Asian children. Second, in terms of cultural adaptation, more Asian children were born to foreign-born parents and spoke a language other than English at home than Latine children or Black children in this sample. In terms of family structure, Black children were more likely to come from single-parent households than Latine children or Asian children. Table 1 shows the bivariate correlations between social position, family structure, parent academic socialization (e.g., family practices, parental expectations, family routines, and leisure activities), WM, and mathematics ability variables. Across all ethnic/racial minority groups, child health, child age, fewer siblings, more close relatives, a mother with a high school diploma or higher, a two-parent home, greater SES, and English spoken in the home were all positively associated with WM and mathematics. Regarding parent academic socialization variables, all showed positive associations with mathematics and WM except for parental warmth, which was not associated with either dependent variable.


*Are parent academic socialization practices predictive of mathematics and WM for a nationally representative sample of minoritized children, after accounting for family position variables and previous WM skills?*


Tests of the path model represented in Figure 1 for the overall sample of ethnic–racial minority children demonstrated a good fit to the data (*Χ*^2^ (1) = 26.890, *p* < 0.0001, CFI = 0.997, RMSEA = 0.062 95%CI [0.043, 0.083], and SRMR = 0.002), suggesting that an exploration of path significance is allowable. The results suggested that parental education expectations (*β* = 0.02; *p* = 0.046), parental racial socialization (*β* = 0.03; *p* = 0.024), and reading activities (*β* = 0.04; *p* = 0.004) were positively predictive of ethnic/racial minority children’s WM. None of the family practices were predictive of mathematics skills. Not surprisingly, WM at T2 was predictive of T3 mathematics abilities (*β* = 0.22; *p <* 0.001) even after controlling for previous mathematics abilities and WM. A summary of unstandardized regressions can be found in Table 2.


*Does WM mediate the relationship between parent academic socialization practices and later mathematics ability for a nationally representative sample of ethnic/racial minority children?*


Tests of the indirect effects of the various parent academic socialization factors on mathematics skills through WM at T2 were conducted with 500 bootstraps. The results supported mediation; thus, WM may serve as a mechanism through which parental education expectations (*β* = 0.005; *p* = 0.048 95%CI [0.000, 0.010]), parental racial socialization (*β* = 0.006; *p* = 0.021 95%CI [0.001, 0.011]), and reading activities (*β* = 0.008; *p* = 0.006 95%CI [0.002, 0.013]) impact mathematics skills.


*Do parents*
*’ academic socialization practices predict mathematical achievement and WM for different ethnic/racial groups?*


Tests of multigroup moderation were conducted to determine whether differences in the prediction of family practices on WM for ethnic/racial minority children exist. These analyses occurred in steps. First, a fully constrained model was tested for the three subgroups of children based on ethnic group membership, forcing all paths to be equal across all groups. This resulted in a good model fit [*Χ*^2^ (98) = 100.698, *p* = 0.4057, CFI= 0.99, RMSEA = 0.003, and SRMR = 0.025]. Next, paths were successively allowed to be freely estimated to determine whether model fit was improved when group differences were allowed to occur. A table of the successive model comparisons of fit (due to releasing specific path constraints), as well as changes in chi-square tests, is shown in Table 3. Releasing constraints on certain path coefficients improved the model fit. Specifically, releasing constraints on the parameters relating regular mealtimes to WM [Δχ^2^(2) = 5.28; *p* = 0.071] and limits on time spent using a screen (e.g., TV) to children’s mathematical skills [Δχ^2^(2) = 5.14; *p* = 0.077] improved model fit. In addition, freeing the constraints on the relations between mathematical skills at T3 and WM at T2 improved model fit [Δχ^2^(2) = 9.02; *p* = 0.011]. As such, we next investigated pairwise between-group differences for those parameters that improved model fit when freed.

### 9.2. Path Relationships by Racial Group

Pairwise comparisons were run for non-invariant parameters. We first looked at differences in the parameter constraints on regular mealtimes predicting WM. An examination of chi-squared difference tests revealed that only the parameter comparison between Latine and Asian children was statistically significant [Δχ^2^(1) = 5.11; *p* = 0.024]; no parameter differences emerged between Black and Asian (*p* = 0.149) or Latine and Black students (*p* = 0.497). Specifically, there was no association between having a regular mealtime and WM for Latine children (*b* = 0.16; *p* = 0.672), though there was a negative association for Asian students (*b* = −1.44; *p* = 0.008).

We next examined pairwise comparisons for the relation between screen time limits and mathematical skills; only a difference between Latine and Black students in the strength of the parameter emerged [Δχ^2^(1) = 5.098; *p* = 0.024]. For Black students, there was a small negative association between setting regular limits on screen use (*β* = −0.05; *p* = 0.041), whereas there was no such association for Latine students (*β* = 0.02; *p* = 0.317).

Finally, we tested pairwise comparisons for the relation between T3 mathematical skills and T2 WM. Chi-squared difference tests revealed differences in the parameter between Latine and Asian students [Δχ^2^(1) = 7.60; *p* = 0.006] and between Latine and Black students [Δχ^2^(1) = 4.05; *p* = 0.044]. In both cases, the association between mathematics and WM was stronger for Latine students (*β* = 0.23; *p* < 0.001) than for either Black students (*β* = 0.19; *p* < 0.001) or Asian students (*β* = 0.15; *p* < 0.001). Over and above associations between T2 working memory, differences emerged between T3 mathematics and T1 WM, though only for Latine and Asian students [Δχ^2^(1) = 3.79; *p* = 0.051]. There was an association between T1 WM and T3 mathematics for Latine students (*β* = 0.08; *p* < 0.001) but not for Asian students (*β* = 0.03, *p* = 0.185).

## 10. Discussion

Despite this focus on the adaptive ways in which families of ethnic minority children support their children’s development, no such work exists as this relates to cognitive development or the development of WM and mathematics skills of this population. By using social capital theory and a community cultural wealth perspective in service of avoiding deficit interpretations and alternatively focusing on familial capital for historically minoritized groups in the U.S., the current study examined the role of family academic socialization strategies in predicting WM and subsequent mathematics for Asian, Black, and Latine children while also exploring the role of social position and family structure. The results support the idea that social position and family structure variables are noteworthy considerations for ethnic/racially minoritized youth. Family SES and English spoken in the home positively predicted mathematics skills for our socio-demographically diverse sample.

Although multigroup moderation based on ethnic/racial group membership was tested, few differences were found. Regarding WM, only the relative strength of the predictive power of having regularity in mealtimes differed by ethnic/racial group. Here, the only difference that emerged was between Latine and Asian children in the sample: while there was no association between the number of regular mealtimes per week for Latine students, there was a—perhaps counterintuitive—negative association for Asian students. In predicting mathematical skill at the end of first grade, racial/ethnic group differences were detected for screen limit setting. Further analysis revealed that only Black students were affected by screen limit setting, though the relation was weak. In both cases, it is important to note that any significant parameter differences were quite small and questionably meaningful, as only one pairwise comparison was significant. That is, in the case of the relation between consistent mealtimes and WM, the parameter differed between Latine and Asian students, though they were statistically indistinguishable between Latine and Black students and between Black and Asian students. The same pattern of findings was described among electronic screen limit setting (though a difference emerged between Latine and Black students). This is all to note, particularly given that both findings were somewhat counterintuitive (increased consistency and limit setting related to worse outcomes), that one group did not differ significantly from the other two. These findings should be interpreted with caution.

While there was a moderate relation between constructs for all ethnic/racial groups—wherein a 0.15 SD increase in mathematical skill corresponded to at least 1 SD increase for all groups—the relation was particularly strong for Latine students, for whom a 1 SD increase in WM was associated with a 0.23 SD increase in mathematical skill. This finding may have little to do with Latine children as an ethnic group and more to do with factors that were not measured herein. These factors might include *bien educado*, as previously noted, encouraging children to have good manners and requiring self-regulation skills, in addition to specific discussions about resilience, familism, and racial socialization messages. The finding that WM and math are related is consistent with prior work [64], demonstrating a strong significant relationship between WM and mathematics skills. It is possible that WM predicts mathematics skills because WM may be related to kindergarten-entry English oral language skills and SES. Indeed, some work suggests that English oral language skills and SES often set up young Latine youth for achievement [65] even when controlling for earlier skills. Here, we did not control for how skilled at English children were. There might also be opportunities to better understand family cultural values and cognitive processes as they relate to early math by examining differences not based on just ethnicity/race but group similarities based on traditions as well [23]. Thus, this assumption awaits further confirmation.

Overall, when predicting school readiness across the three groups of children, parental education expectations, parental racial socialization, and reading activities were positively associated with children’s WM. Of the family routines in the model, as opposed to specific family academic socialization practices, only having an established time limit on television was predictive of mathematics skills. We also found that WM mediated the relationship between parent academic socialization practices and mathematics skills, thus highlighting the importance of better understanding WM skills and how they are developed for these groups of children. The results also indicated that WM mediated the associations of parental education expectations, parental racial socialization, and reading activities with mathematics skills. This suggests that the positive influences of parental socialization, education expectations, and reading on children’s mathematics abilities can be explained by their associations with WM. Consistent with other work, it may be the case that some parental racial socialization messages are related to how ethnic–racial minority children should approach schooling as a buffer against discrimination, which may direct children to learn how to thrive in school. Of course, more work about how these questionnaires may be differentially interpreted is required to confirm these assertions. Further, the relations between parenting practices and WM may offer clues on how parents may prepare their children for schooling in the domain of general areas. It may be the case that when parents socialize their children to engage in behaviors that support their broader self-regulation, this also predicts domain-specific skills. This is potentially a future direction in this area.

Further, the current study supports the contentions of prior work that suggest that WM mediates the relationship between academic routines (e.g., instruction) and mathematics skills [66]. Indeed, children with stronger WM demonstrated greater benefits from parental academic socialization practices for mathematical outcomes than children with less developed WM. Moreover, prior work also suggests the importance of parents having high educational expectations for their children and reading to children regularly [12,67], as well as parent racial socialization practices [15] in academic outcomes. Similarly, WM also mediates the relationship between parent academic socialization practices and mathematics skills, in that, as parents provide high educational expectations and racial socialization, ethnic–racial children’s WM skills are bolstered. Despite the WM task being a backward digit span, children may be relying more on strategies they have learned, rather than an inherent WM skill, that may be derived from home experiences.

## 11. Limitations

Among the strengths of this study was the use of an asset-based theoretical framework as the lens to understand the results of multigroup analyses meant to examine how family academic socialization practices differ across ethnic/racial minority groups. By using familial capital and this lens for our work, we were able to extend a traditional understanding of the results through a normative perspective by incorporating a culturally sensitive perspective.

Despite these strengths, this study is limited by the variables included in the ECLS-K dataset. There are numerous other variables related to familial capital and community cultural wealth that may help us understand the development of ethnic–racial minority children. More specifically, variables related to racial socialization, more specific family routines, types of family outings, and variables regarding how families use their leisure time with their children would have been informative and more clearly linked to family practices. For instance, we used bedtimes to indicate that routines were in place, although this variable could mean that children have a set bedtime but no other routine in place. Future research might address this by testing hypotheses associated with the integrative model using multiple datasets. This has the potential to offer further support for the model or allow us to consider how we might adjust the model for better predictability. In any case, using large datasets with sizeable samples of ethnic–racial minority children is one route that may ensure further testing for this model and its claims. Further, an opportunity to use earlier ECLS-K cohorts was not possible because earlier versions did not include a WM measure, a focal variable in the current study. Future work should use more relevant variables, as well as cognitive measures like executive functions, to provide a deeper empirical examination of family practices that promote development while attending to differences in how Asian-American, Black, and Latine children and their families experience the constructs articulated in Integrative Theory. This might best be approached by first utilizing qualitative methodology to determine what family practices families use to begin with, particularly when these practices deviate from what the majority population in the U.S. uses. This approach would provide more nuance as to why this was the result.

Next, we observed that only 16% of Asian-American parents were born in the U.S. compared with 89% of Black and 44% of Latine parents in the ECLS-K dataset. This limits our understanding of these familial capital and family processes in many ways. First, it is difficult to disentangle those family processes that may have emerged as a result of migration and assimilation as opposed to simply being an ethnic/racial minority in the U.S., even though they have experienced decades of marginalization. Future research should consider immigration across groups, intergenerational effects, and acculturation as additional possible factors that may contribute to the development of WM in the context of academic skills and, potentially, math skills for these groups.

Further, the only WM measure in the ECLS-K dataset is the numbers reversed task. There are indeed various ways to measure WM, and thus, our findings are specific to children’s performance in the numbers reversed task. We know from prior work that WM is associated with mathematics skills, regardless of task type, though the magnitude of the relationship varies [68]. It is possible that, had we used an alternative WM measure, such as a composite WM task, we would have found stronger correlations based on prior work, whereas, with a verbal WM task, we might have observed smaller relationships [68].

Beyond the limitations when working with a large, secondary dataset, we acknowledge the small effect sizes observed in our study. Furthermore, we feel it is worth highlighting a limitation in that these findings may not be generalizable beyond the unique context of the U.S., where the classroom language of instruction is nearly always and exclusively English, where there are deep-seated biases against non-White families rooted in a foundation of White supremacy and where school funding and access to resources is typically highly correlated with family socioeconomic status. Finally, it is important to note that, while the groups included in this analysis represent broad swathes of the population, it is likely there is as much within-group variability as there is between groups. Further investigation is necessary to better understand how subpopulations have collapsed under ethnically/racially minoritized groups (here, Black/African-American, Asian/Asian-American, Latine) differ. Due in part to all these limitations, these findings should be interpreted with caution, as they cannot yet help us make contributions to our understanding of applying Integrative Theory to cognitive development. These findings are indeed preliminary, but we urge other scholars to think critically about using theory in cognitive development as it relates to minoritized children.

## 12. Conclusions

Perhaps the most important lesson to be learned from this study is that the measures and constructs previously created and operationalized for all children may not have much predictive utility when making claims about ethnic/racial minority children. Predictions regarding WM and mathematics skills have often been conceptualized and interpreted based on prior work that included White children from affluent families. By using an asset-based lens, we can begin to understand the results not only from a normative perspective but also from a culturally sensitive perspective. Instead of assuming the constructs we previously investigated are operationalized and utilized similarly across all groups, we must further interrogate these measures since they were developed based on White population norms. By scrutinizing our methods of assessing and interpreting the development of children, we create opportunities for equity. Researchers must continue to question whether they are assessing accurate depictions of children and families’ lives, especially across cultural lines.

## Figures and Tables

**Figure 1 behavsci-14-00390-f001:**
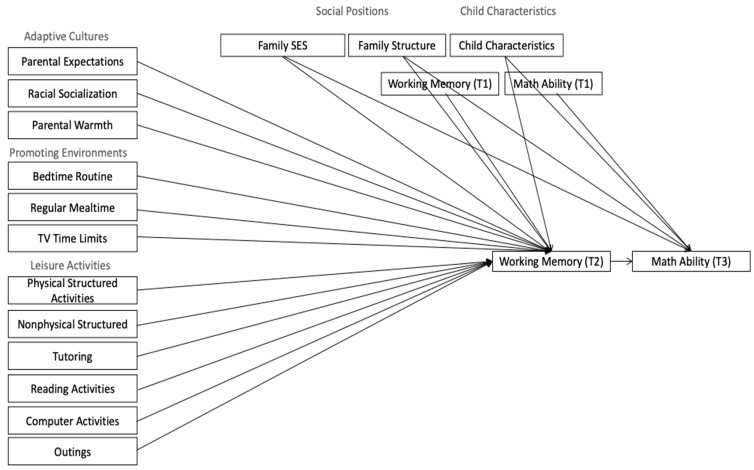
Proposed model for predicting working memory and mathematics abilities. Direct relationships between family practices and math abilities are not shown to increase the readability of the figure.

**Table 1 behavsci-14-00390-t001:** Descriptive statistics and correlations for the overall sample.

	1	2	3	4	5	6	7	8	9	10	11	12	13	14	15	16	17	18	19	20	21	22	23
1. SES	-																						
2. Language	0.29 ***	-																					
3. Marital	0.34 ***	−0.06	-																				
4. Education	0.51 ***	0.35 ***	0.12 ***	-																			
5. Grandparents	−0.23 ***	0.21 ***	0.02	0.17 ***	-																		
6. Siblings	−0.21 ***	−0.07 *	0.04	−0.22 ***	−0.08 *	-																	
7. Gender	−0.01	0.01	0.05	−0.02	−0.06	−0.01	-																
8. Age	0.00	0.05	−0.03	0.02	0.01	0.03	−0.08 *	-															
9. Health	0.19 ***	0.20 ***	0.00	0.17 ***	0.13 ***	−0.05	0.04	−0.01	-														
10. Expectations	0.08 *	−0.12 ***	0.05	0.50	−0.04	−0.03	0.06	0.01	0.00	-													
11. Racial Soc.	0.10 **	0.07 *	0.02	0.11 **	−0.01	−0.06	0.01	0.00	0.09 **	0.07	-												
12. Warmth	0.04	0.08 *	0.03	0.02	0.00	−0.08 *	−0.04	0.01	0.06	0.03	0.03	-											
13. Bedtime	0.05	0.13 ***	0.05	0.01	0.08 *	0.01	−0.03	−0.01	0.08 *	0.00	0.04	0.02	-										
14. Mealtime	0.12 ***	0.07	0.08 *	0.02	0.02	0.11 **	−0.04	−0.03	0.08 *	0.02	0.07 *	0.06	0.17 ***	-									
15. TV Limits	0.09 *	−0.03	0.04	−0.01	0.01	−0.06	−0.05	−0.06	0.01	0.06	0.16 ***	0.05	0.10 **	0.1 7***	-								
16. Physical	0.36 ***	0.18 ***	0.09 **	0.20 ***	0.20 ***	−0.12 ***	0.06	0.00	0.05	0.04	0.11 **	0.04	0.12 **	0.08 *	0.09 *	-							
17. Nonphysical	0.23 ***	0.10 **	0.12 ***	0.16 ***	0.09 **	−0.11 **	0.11 **	−0.01	0.05	0.05	0.15 ***	0.06	0.05	0.08 *	0.01 **	0.38 ***	-						
18. Service	0.13 ***	0.10 **	0.09 *	0.09 *	0.04	−0.03	0.03	0.00	0.01	0.06	0.14 ***	0.07 *	0.07 *	0.13 ***	0.13 ***	0.15 ***	0.33 ***	-					
19. Computer	0.03	0.12 **	0.00	0.05	0.06	0.00	0.03	0.06	0.02	−0.02	0.06	0.03	0.05	0.03	−0.01	0.06	0.10 **	0.05	-				
20. Outings	0.23 ***	0.20 ***	0.04	0.18 ***	0.09 **	−0.06	0.05	−0.02	0.13 ***	0.07 *	0.17 ***	0.06	0.13 ***	0.18 ***	0.16 ***	0.29 ***	0.27 ***	0.13 ***	0.11 **	-			
21. WM T1	0.37 ***	0.16 ***	0.14 ***	0.22 ***	0.10 **	−0.07	−0.04	0.06	0.10 **	0.02	0.09 *	0.05	0.05	0.05	0.05	0.17 ***	0.09 *	0.04	0.05	0.08 *	-		
22. WM T2	0.31 ***	0.17 ***	0.12 ***	0.23 ***	0.11 **	−0.11	0.03	0.08 *	0.15 ***	0.10 **	0.11 **	0.02	0.09 *	0.11 **	0.02	0.16 ***	0.13 ***	0.06	0.03	0.10 **	0.55 ***	-	
23. Math T3	0.43 ***	0.12 **	0.20 ***	0.23 ***	0.12 ***	−0.14	−0.03	0.11 **	0.17 ***	0.07 *	0.10 **	0.00	0.07 *	0.12 **	0.10 **	0.19 ***	0.14 ***	0.08 *	0.07 *	0.14 ***	0.56 ***	0.55 ***	--
Mean	−0.30	0.65	0.53	0.77	2.01	1.52	0.50	72.58	3.27	5.53	3.07	3.72	0.72	4.43	0.70	0.49	0.52	0.59	2.21	2.24	425.75	443.12	52.39
Standard Dev.	0.82	0.48	0.50	0.42	1.24	1.19	0.50	4.09	0.88	1.27	1.33	0.59	0.45	1.65	0.46	0.63	0.93	0.61	0.82	1.41	29.10	31.33	14.43
Skewedness	0.67	−0.61	−0.13	−1.31	0.25	1.16	0.01	0.20	−0.96	−0.80	−0.18	−2.18	−0.95	−0.20	−0.90	0.91	2.12	0.53	−0.75	0.10	0.80	−0.01	0.55
Kurtosis	0.04	−1.62	−1.98	−0.28	−0.57	2.87	−2.00	0.20	−0.04	0.74	−1.07	4.46	−1.09	−0.36	−1.19	−0.22	4.70	−0.62	−0.18	−0.81	−0.64	−1.19	0.54

NOTE: *** *p* < 0.001, ** *p* < 0.01, and * *p* < 0.05; SES—socioeconomic status; Racial Soc.—racial socialization; WM—working memory.

**Table 2 behavsci-14-00390-t002:** Unstandardized direct and indirect effects of variables in the model for the overall sample.

	Working Mem (T2)	Math (T3)	Math ^a^
Predictors			
**Parental Expectations**			
Education Expectations	0.566 *	0.077	0.062 *
Racial Socialization	0.653 *	0.001	0.072 *
Parental Warmth	0.166	0.040	0.018
**Family Routines**			
Bedtime	0.412	0.483	0.046
Regular Mealtime	−0.157	0.091	−0.017
TV Time limits	−1.240	−0.352	−0.137
**Leisure Activities**			
Physical Structured	0.584	0.164	0.064
Nonphysical Structured	−0.358	−0.042	−0.040
Tutoring	0.843	0.056	0.094
Reading Activities	1.485 **	0.266	0.164 **
Computer Activities	−0.023	−0.006	−0.003
Outings	−0.199	0.300 *	−0.022
Working Mem (T1)	0.350 ***	----	
Math Skills (T1)	0.909 ***	0.829 ***	
Working Mem (T2)	--	0.110 ***	

* *p* < 0.05. ** *p* < 0.01. *** *p* < 0.001. ^a^ This refers to the indirect effect of parent academic socialization on math through working memory at T2.

**Table 3 behavsci-14-00390-t003:** Model fit statistics for all models.

Model	*Χ*^2^(df)	Δ*Χ*^2^	Δ*Χ*^2^ *p*-Value	Invariant?
Fully Constrained	100.698(98)			
Working Memory				
**Parental Expectations**				
Education Expectations	99.725	0.97	0.615	Yes
Racial Socialization	96.662	4.04	0.133	Yes
Parental Warmth	98.851	1.85	0.397	Yes
**Family Routines**				
Bedtime	97.647	3.05	0.218	Yes
Regular Mealtime	95.419	5.28	0.071	No
TV Time Limits	99.81	0.89	0.641	Yes
**Leisure Activities**				
Physical Structured	99.743	0.95	0.620	Yes
Nonphysical Structured	97.108	3.59	0.166	Yes
Tutoring	99.988	0.71	0.701	Yes
Reading Activities	96.247	4.45	0.108	Yes
Computer Activities	98.059	2.64	0.267	Yes
Outings	100.65	0.05	0.976	Yes
Mathematics Skills				
**Parental Expectations**				
Education Expectations	97.572	3.13	0.210	Yes
Racial Socialization	95.563	5.13	0.077	Yes
Parental Warmth	97.349	3.35	0.187	Yes
**Family Routines**				
Bedtime	99.445	1.25	0.534	Yes
Regular Mealtime	101.125	−0.43	1.000	Yes
TV Time Limits	94.828	5.87	0.053	No
**Leisure Activities**				
Physical Structured	97.991	2.71	0.258	Yes
Nonphysical Structured	98.679	2.02	0.364	Yes
Tutoring	97.028	3.67	0.160	Yes
Reading Activities	100.859	−0.16	1.000	Yes
Computer Activities	100.739	−0.04	1.000	Yes
Outings	99.313	1.38	0.500	Yes
Working Memory	91.682	9.02	0.011	No

## Data Availability

The data are available on https://nces.ed.gov and were accessed on 7 May 2019.

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
