# Peer review of "Understanding Working Memory and Mathematics Development in Ethnically/Racially Minoritized Children through an Integrative Theory Lens"

_behavsci, 2024, doi:10.3390/bs14050390_

Round 1

Reviewer 1 Report

Comments and Suggestions for Authors

Thank you for the opportunity to read this manuscript. The paper provides great insights of the current study and sets up potentials for further research. I have very minor suggestions for the authors.

Page 1, lines 35-38: This sentence reads as thought Coll et al, and Perez-Brena are stating that minority children are lacking in comparison to white students. However, Coll and Perez-Brena have argued otherwise. It would be helpful to rephrase this sentence to make it more clear for reading audiences.

Page 2, line 46, minor details but very important for the scholar. You are missing the accent on her name.

Page 2, lines 46-48, I think the use of interdisciplinary approaches is great and allows for a more nuanced understanding of the developmental processes in minoritized children. I’m glad to see these three frameworks pulled together as framing and lens.

Page2, lines 63-65, recent research by Beltrán-Grimm, 2022 (Latina Mothers' Cultural Experiences, Beliefs, and Attitudes May Influence Children's Math Learning) also point to socio-political and cultural nuances in math.

Multi-Theoretical Framework. Great comprehension of theories, emphasizing on cultural and family contexts.

Page 4, lines 159-165. I would suggest to also mention that Latine family obligation, in this case familismo, while an important role in the Latine culture, it also has limitation and can become a negative factor. Ex, language brokering is for many a Latine family obligation that can have both positive and negative outcomes.

Page 4, lines 82-83, while Halgunseth is the correct scholar to cite, if you are integrating interdisciplinary frameworks, (Valdés, 1996), is the scholar who brought these concepts to the front.

Family Structure and Connectedness; Family Routines and Leisure Activities;

Parental Warmth – good sections!

Page 5, lines 42-43, this is where I recommend, Valdés, 1996, and other scholars similar work that states that for Latine parents being “bien educado” is for their children to behave and have good manners, which inherently is self-regulation.

Family Practices, Working Memory, and Mathematics – good section!

Analytic Approach: Because the authors are intentional on the cultural lens and framing approach, please add why using this model choice in relation to the theoretical cultural framing the authors are using.

Page 11, results: This is interesting, none of the family practices examined were predictive of mathematics skills, they do not directly translate to mathematics ability in this model, curious if a qualitative methodology for this would provide more nuance as to why this was the result.

Page 14, I’m curious if the reading activates in Latine familiar were bilingual or in Spanish.

Discussion: Interesting how the stronger relationship between WM and math skills among Latine children compared to other groups suggests that there might be unmeasured factors influencing this association, which goes back to your cultural framing and lens. There are potential underlying factors, but what are those factors??? Very interesting finding.

Author Response

Thank you for the opportunity to read this manuscript. The paper provides great insights of the current study and sets up potentials for further research. I have very minor suggestions for the authors.

  • Thank you for the compliment!

Page 1, lines 35-38: This sentence reads as thought Coll et al, and Perez-Brena are stating that minority children are lacking in comparison to white students. However, Coll and Perez-Brena have argued otherwise. It would be helpful to rephrase this sentence to make it more clear for reading audiences.

  • Thank you for highlighting this very important point. We have rephrased this to accurately reflect this.

Page 2, line 46, minor details but very important for the scholar. You are missing the accent on her name.

  • We have updated this very important detail.

Page 2, lines 46-48, I think the use of interdisciplinary approaches is great and allows for a more nuanced understanding of the developmental processes in minoritized children. I’m glad to see these three frameworks pulled together as framing and lens.

  • Thank you very much for this kind remark!

Page 2, lines 63-65, recent research by Beltrán-Grimm, 2022 (Latina Mothers' Cultural Experiences, Beliefs, and Attitudes May Influence Children's Math Learning) also point to socio-political and cultural nuances in math.

  • This is important work to integrate into our paper. We have added this reference on p. 2.

Multi-Theoretical Framework. Great comprehension of theories, emphasizing on cultural and family contexts.

  • Thank you!

Page 4, lines 159-165. I would suggest to also mention that Latine family obligation, in this case familismo, while an important role in the Latine culture, it also has limitation and can become a negative factor. Ex, language brokering is for many a Latine family obligation that can have both positive and negative outcomes.

  • This is a great point. We have added this language to p. 4. Thank you for pointing this out.

Page 4, lines 82-83, while Halgunseth is the correct scholar to cite, if you are integrating interdisciplinary frameworks, (Valdés, 1996), is the scholar who brought these concepts to the front.

  • Thank you bringing this to our attention. We had added this work on p. 4 and we believe it reads much better as a result.

Family Structure and Connectedness; Family Routines and Leisure Activities;

Parental Warmth – good sections!

  • Thank you!

Page 5, lines 42-43, this is where I recommend, Valdés, 1996, and other scholars similar work that states that for Latine parents being “bien educado” is for their children to behave and have good manners, which inherently is self-regulation.

  • What a great point! We have added this language to the paper and believe it adds a lot of depth to our argument.

Family Practices, Working Memory, and Mathematics – good section!

  • Thank you!

Analytic Approach: Because the authors are intentional on the cultural lens and framing approach, please add why using this model choice in relation to the theoretical cultural framing the authors are using.

  • Thank you for the opportunity to justify the analytic model against the theoretical framing. We have added language to this section to provide rationale for how we mapped these variables onto our analyses.

Page 11, results: This is interesting, none of the family practices examined were predictive of mathematics skills, they do not directly translate to mathematics ability in this model, curious if a qualitative methodology for this would provide more nuance as to why this was the result.

  • We agree that it is odd that none of the family practices were predictive of math skills. We believe there is more work necessary for this field to understand family practices, especially ones that deviate from a White Eurocentric perspective. We’ve added this point as a limitation on p. 18.

Page 14, I’m curious if the reading activates in Latine familiar were bilingual or in Spanish.

  • We imagine so, but have no information to say in either direction.

Discussion: Interesting how the stronger relationship between WM and math skills among Latine children compared to other groups suggests that there might be unmeasured factors influencing this association, which goes back to your cultural framing and lens. There are potential underlying factors, but what are those factors??? Very interesting finding.

  • We agree that it is an interesting finding. We believe there is more work necessary for this field to understand family practices, especially ones that deviate from a White Eurocentric perspective. We’ve added this point as a limitation.

Reviewer 2 Report

Comments and Suggestions for Authors

Congratulations to the authors because the work done is impressive. It concerns a very important topic, because taking into account culture to study executive functions is something that has been missed during a long time in the field. The manuscript is well written, the study is sound and the conclusions are well derived from the data. I have some minor comments:

  1. I suggest integrating the introduction and discussion sections within the theoretical context proposed by Gaskins & Alcalá (2023). 10.1080/15248372.2022.2160722
  2. The fit indexes showed of the initial model (Figure 1) seems too good for such a complex model. Did the authors include any correlation among variables in the same Time? For example, is hard to imagine that reading activities are not correlated to TV Time limits. If the authors included some correlations or if they used any modification index to improve the model, please, report them.
  3. In addition, try to explain better each variable included in the model. For example, in the Instruments section, the authors explain that they asked for frequency playing games and puzzles with child. I imagine that this question is included in the Nonphysical Structured leasure activities, but it is not mentioned in any place. So, please, explain in detail every variable included in the model.

Author Response

Congratulations to the authors because the work done is impressive. It concerns a very important topic, because taking into account culture to study executive functions is something that has been missed during a long time in the field. The manuscript is well written, the study is sound and the conclusions are well derived from the data.

  • Thank you for this kind remark!

I suggest integrating the introduction and discussion sections within the theoretical context proposed by Gaskins & Alcalá (2023). 10.1080/15248372.2022.2160722

  • Great point! We have now added more work from Gaskins & Alcalá (2023) in the Introduction and Discussion questions.

The fit indexes showed of the initial model (Figure 1) seems too good for such a complex model. Did the authors include any correlation among variables in the same Time? For example, is hard to imagine that reading activities are not correlated to TV Time limits. If the authors included some correlations or if they used any modification index to improve the model, please, report them.

  • All exogenous variables were correlated for the purposes of invoking FIML in the overall path analysis. This is reflected in the model fit and the fact that this is a nearly fully-identified model (DF = 1). For the purposes of clarity and interpretability, we do not include pathways signifying correlations among exogenous variables. We have included a statement to this effect in the analytic plan.

In addition, try to explain better each variable included in the model. For example, in the Instruments section, the authors explain that they asked for frequency playing games and puzzles with child. I imagine that this question is included in the Nonphysical Structured leasure activities, but it is not mentioned in any place. So, please, explain in detail every variable included in the model.

  • Thank you for bringing our attention to this very confusing section. We have now revised it to better map on to the figure and give specific details about each variable in the model.